# A Novel Strategy for the Synthesis of High Stability of Luminescent Zero Dimensional–Two Dimensional CsPbBr_3_ Quantum Dot/1,4-bis(4-methylstyryl)benzene Nanoplate Heterostructures at an Atmospheric Condition

**DOI:** 10.3390/nano13192723

**Published:** 2023-10-07

**Authors:** Yanran Wang, Ming-yu Li, Shijie Liu, Yuan Ma, Bo Sun, Liangyu Wang, Haifei Lu, Xiaoyan Wen, Sisi Liu, Xumin Ding

**Affiliations:** 1Donghai Laboratory, Zhoushan 316021, China; 312506@whut.edu.cn; 2School of Science, Wuhan University of Technology, Wuhan 430070, China; liusj@whut.edu.cn (S.L.); 327402@whut.edu.cn (Y.M.); 285422@whut.edu.cn (B.S.); 333616@whut.edu.cn (L.W.); haifeilv@whut.edu.cn (H.L.); wenxy@whut.edu.cn (X.W.); 3Advanced Microscopy and Instrumentation Research Center, School of Instrumentation Science and Engineering, Harbin Institute of Technology, Harbin 150090, China; xuminding@hit.edu.cn

**Keywords:** perovskite quantum dots, *p*-MSB nanoplate, typical type-II heterostructure, high photoluminescence quantum yield, moisture stability

## Abstract

Perovskite quantum dots (QDs), emerging with excellent bright-green photoluminescence (PL) and a large absorption coefficient, are of great potential for the fabrication of light sources in underwater optical wireless communication systems. However, the instability caused by low formation energy and abundant surface traps is still a major concern for perovskite-based light sources in underwater conditions. Herein, we propose ultra-stable zero dimensional–two dimensional (0D–2D) CsPbBr_3_ QD/1,4-bis(4-methylstyryl)benzene (*p*-MSB) nanoplate (NP) heterostructures synthesized via a facile approach at room temperature in air. CsPbBr_3_ QDs can naturally nucleate on the *p*-MSB NP toluene solution, and the radiative combination is drastically intensified owing to the electron transfer within the typical type-II heterostructures, leading to a sharply increased PLQY of the heterostructure thin films up to 200% compared with the pristine sample. The passivation of defects within CsPbBr_3_ QDs can be effectively realized with the existence of *p*-MSB NPs, and thus the obviously improved PL is steadily witnessed in an ambient atmosphere and thermal environment. Meanwhile, the enhanced humidity stability and a peak EQE of 9.67% suggests a synergetic strategy for concurrently addressing the knotty problems on unsatisfied luminous efficiency and stability of perovskites for high-performance green-emitting optoelectronic devices in underwater applications.

## 1. Introduction

In recent years, the fabrication of high-performance light-emitting diodes (LEDs) and laser diodes (LDs) has drawn ever-growing research interests as light sources for underwater optical wireless communication (UOWC) systems [1] owing to their diverse potential applications in submarine communications, deep ocean exploration and underwater wireless sensor networks (UWSNs), etc. [2,3,4]. Recently, metal halide perovskites (MHPs) have been regarded as promising solid-state light-emission materials in LEDs and gain mediums in LDs because of their excellent photoluminescence quantum yield (PLQY), superior exciton binding energy, narrow emission, and tunable bandgap [5,6]. Among MHPs, cesium lead halide quantum dots (CsPbBr_3_ QDs) exhibit excellent scintillation properties [7,8], outstanding bright-green photoluminescence (PL), and high PLQY [9], which have been witnessed near-unity PLQYs in solution [10]. However, the agglomeration of nanocrystals during the fabrication of thin films can potentially result in the severe deterioration of PLQY, down to 18% [11], in association with loss of quantum confinement [12], and thus a feasible strategy on the realization of highly luminescent thin films with perovskite QDs is still a crucial problem for application in light-emitting devices [13]. Providing the low formation energy [14] and mobile ionic structure with abundant surface traps [15], perovskites are generally vulnerable to external factors, including oxygen, temperature, and moisture [16], inherently hindering the development of high-performance underwater light sources [17].

To date, numerous researches have increasingly endeavored to explore an effective approach for improving the stability of CsPbBr_3_ QDs [14], and a popular routine strategy is to construct relatively inert shells or barrier matrixes among the QDs, preventing the undesirable erosion of oxygen and moisture [18]. For instance, the uniform CsPbBr_3_@CdS core/shell QDs were synthesized with significantly improved stability under 75% relative humidity (RH) at the expense of the slightly decreased PLQY for the colloidal QD solution from 90% to 88% [19]. Meanwhile, the mono-dispersed CsPbBr_3_@SiO_2_ colloidal QDs were synthesized at −20 °C with the enhanced PLQY for the solution of blue emission at 460 nm from 72.4% to 75.6% [20]. Unfortunately, the surface passivation with inorganic oxides was inevitably porous due to the lattice mismatch between CsPbBr_3_ and SiO_2_ [21], which can suffer a high permeable possibility of external H_2_O/O_2_ [22]. To minimize the lattice mismatch between foreign materials, CsPbBr_x_ shells were grown in situ on CsPbBr_3_ QDs via a facile hot injection method, and the blue emission PLQY of CsPbBr_3_@ Amorphous-CsPbBr_x_ QD solution was radically increased from 54% to 84% in comparison with pristine cubic CsPbBr_3_ QD solution under excitation of 365 nm light illumination [23]. Despite a noticeably improved PLQY, the decomposition of the CsPbBr_x_ still remains a big concern under exposure to moisture and oxygen [22,24]. Additionally, perovskite quantum dots can also be packed into crosslinked polystyrene (PS) beads via a simple swelling–shrinking strategy, which exhibited an extradentary water-resistant property [25]. Meanwhile, the construction of non-luminescent polymer matrixes can somewhat sacrifice effective area, in turn leading to undesirable irradiation attenuation [26]. Therefore, a well-balanced approach between long-term stability and high PLQY is currently a critical obstacle for the CsPbBr_3_ QDs as light-emission materials of light sources in particle applications [13,27].

Here, we demonstrate a synthetic strategy to simultaneously enhance the PLQY and stability under various conditions with a novel zero dimensional–two dimensional (0D–2D) CsPbBr_3_ quantum dot (QD)/1,4-bis(4-methylstyryl)benzene (*p*-MSB) nanoplate (NP) heterostructures. The 0D–2D heterostructures were facilely synthesized via spontaneous nucleation of CsPbBr_3_ QDs on *p*-MSB nanoplates at room temperature under an atmosphere condition, and the PLQY of the resulting thin films radically increased by 200% with an optimized blending concentration of *p*-MSB at 1mg/mL owing to the photoactive carrier transfer from *p*-MSB NPs to CsPbBr_3_ QDs. Benefiting from the hydrophobicity of *p*-MSB, the heterostructure thin films constantly exhibits an outstanding stability under various temperatures throughout the whole testing duration, which is theoretically discussed with the density functional theory (DFT) calculations. Eventually, the 0D–2D CsPbBr_3_ QD/*p*-MSB heterostructures are engaged for the fabrication of green-emitting LEDs with an external quantum efficiency (EQE) of up to 9.67%.

## 2. Materials and Methods

Chemicals: Cesium carbonate (Cs_2_CO_3_, 99%, Aladdin, Riverside, CA, USA), propionic acid (C_3_H_6_O_2_ ≥ 99.5%, SCRC), lead(II) bromide (PbBr_2_, 99.99%, Aladdin), butylamine (C_4_H_11_N, 99.5%, Aladdin), isopropanol (C_3_H_8_O ≥ 99.7%, SCRC), 1,4-Bis(4-methylstyryl)benzene (C_24_H_22_ > 98%, TCI) (*p*-MSB), n-hexane (C_6_H_14_ ≥ 97%, SCRC), and toluene (C_7_H_8_ ≥ 99.5%, SCRC).

Preparation of hybrid CsPbBr_3_ QD/*p*-MSB nanoplates solutions: The Cs^+^ precursor was initially prepared by dissolving 0.5868 g Cs_2_CO_3_ powders into 1 mL propionic acid under ultra-sonification at room temperature, and 25 μL Cs^+^ precursor was subsequently injected into a mixture of 5 mL isopropanol and 10 mL n-hexane, followed by vigorous stirring for 2 min. Meanwhile, 0.9175 g PbBr_2_ powders were dissolved in a 5 mL mixed solution of butylamine, isopropanol, and propionic acid with a volume ratio of 1:1:1. Finally, CsPbBr_3_ QDs were synthesized by swiftly adding 0.27 mL PbBr_2_ solution into the as-prepared Cs^+^ precursor under ambient conditions and were dispersed in the 8 mL toluene solution with various *p*-MSB concentrations between 0 and 2 mg/mL after a centrifugation process at 2000 r/min for two minutes, respectively.

Deposition of the hybrid thin films: Prior to the deposition, each glass substrate with a size of 2.5 × 2.5 cm^2^ was cleaned via a series of sonicate treatments in deionized (DI) water, ethanol, and acetone for 20 min. After drying with nitrogen, the CsPbBr_3_ QD/*p*-MSB nanoplate hybrid thin films were deposited by a sequential spin-coating process on glass substrates at 500 rpm for 20 s and 2000 rpm for 40 s, and the thin films were immediately treated with an annealing procedure at 80 °C for 1 min to evaporate the solvent and remove organic residuals. The devices were fabricated after 3 depositions, and then were annealed at 80 °C for 5 min to guarantee crystallization.

Device characterization: The transmission electron microscope (TEM, FEI Tecnai G2 F20, FEI Co., Hillsboro, OR, USA) was employed for topological characterization of hybrid nanostructures. The surface morphologies of the resulting samples were characterized with a scanning electron microscope (SEM, Hitachi, Tokyo, Japan), and the corresponding elementary analysis was performed with an integrated energy-dispersive spectroscopy system (EDS, Regulus 8100, Hitachi, Tokyo, Japan). The room temperature photoluminescence (PL) and UV-Vis absorption spectra were recorded with a Shimadzu RF-6000 Spectro fluorophotometer (Kyoto, Japan) and a Shimadzu UV-2600 spectrophotometer (Kyoto, Japan) with an excitation light of 350 nm, respectively. Fluorescence quantum yields (PLQY) (relative values) of samples were calculated according to the following expression:(1)$$Yu=Ys\times \frac{Fu}{Fs}\times \frac{As}{Au}$$
where *Ys* is the fluorescence quantum yield to the reference, *Fs* and *Fu* are the integral fluorescence intensity of the reference and sample, and *As* and *Au* are the absorbance at the excitation wavelength of the sample and reference, respectively. The reference materials selected for the PLQY calculation of solutions and films were rhodamine B (quantum yield 89%) and standard sample (quantum yield 55%). The surface roughness conformed with an atomic force microscopy (AFM, Dimension Icon, Bruker, Karlsruhe, Germany), and X-ray diffraction (XRD) measurements were carried out by a Japan Rigaku Smartlab SE diffractometer with a Cu Kα radiation source. Raman spectra were obtained at room temperature using a LabRAM HR Evolution Horiba spectrometer. The time-resolved PL (TRPL) spectra were obtained with a spectrophotometer (Edinburgh FLS1000, Edinburgh Company, Edinburgh, UK), and an X-ray photoelectron instrument (ESCALAB Xi+, Thermo Fisher Scientific, Waltham, MA, USA) was used for the X-ray photoelectron spectroscopy (XPS) images for each sample. Supersaturated salt solutions of LiF, NaCl, and KCl (25 °C) in a sealed chamber were employed to achieve a relative humidity (RH) level of 30%, 75%, and 84%, respectively.

Water molecule adsorption simulation: The absorption energy for water molecules on the surface of the hybrid CsPbBr_3_ QD/*p*-MSB nanoplates was calculated based on the density functional theory (DFT). The exchange–correlation between electrons was described by the generalized gradient approximation (GGA) in the Perdew–Burke–Ernzerhof (PBE) form. A cut-off energy of 300 eV was chosen for the plane-wave basis set in all calculations. The relaxation of geometry optimization was continued until the residual force was less than 0.05 eV/A and the total energy was lower than 2 × 10^−5^ eV per atom. The Monkhorst–Pack k-point mesh of 5 × 5 × 2 was used for the geometry optimizations and static electronic structure calculations. A vacuum slab of 20 Å was introduced to provide enough distance for the hindrance of interactions between layers.

## 3. Results and Discussion

Figure 1a illustrates the fabrication process of zero dimensional–two dimensional (0D–2D) CsPbBr_3_ quantum dot (QD)/*p*-MSB nanoplate (NP) thin films, and the corresponding morphologies of the CsPbBr_3_ QD/*p*-MSB NP heterostructures are shown in Figure 1b–d. To be specific, CsPbBr_3_ QDs were simply obtained by a liquid-phase reaction between PbBr_2_ and Cs^+^ precursor solutions at room temperature under atmospheric conditions, and the as-synthesized CsPbBr_3_ QDs were blended with *p*-MSB NPs with a variation of concentrations between 0 and 2 mg/mL, as depicted in Figure 1a. As shown in Figure 1b, the CsPbBr_3_ QDs uniformly decorated on the surface of *p*-MSB NPs. As evidenced with clear lattice fringes shown in Figure 1c,d, the resulting CsPbBr_3_ QDs on *p*-MSB NPs exhibited excellent crystallinity [28], and the interplanar distance of 4.2 Å can be assigned to the (110) crystal plane of the cubic perovskite [29]. Appendix A shows the typical diffraction rings corresponding to the (001), (011), and (012) planes [30,31] within the perovskite structure, suggesting a well-defined crystalline structure of CsPbBr_3_ QDs on *p*-MSB NPs [32].

To investigate the evolution of optical properties, a series of characterizations was applied for the 0D–2D CsPbBr_3_ QD/*p*-MSB NP solutions depending on the *p*-MSB concentrations: 2 mg/mL (Sp1), 1 mg/mL (Sp2), 0.5 mg/mL (Sp3), 0.1 mg/mL (Sp4), 0.05 mg/mL (Sp5), 0.025 mg/mL (Sp6), 0 mg/mL (Sp7). As shown in Figure 2a, each sample exhibited an identical absorption edge at ~530 nm in the spectra, confirming a negligible effect of the *p*-MSB concentrations on the crystal structure and size distribution of CsPbBr_3_ QDs [33]. With the elevated *p*-MSB concentrations, the absorption curves slightly bent down above 400 nm, which can be inherited from the behavior of the *p*-MSB NPs as evidenced by the absorbance spectrum in Appendix A. As depicted with the Tauc plots in Appendix A, the 0D–2D heterostructures correspondingly maintained a comparable optical bandgap of ~2.35 eV regardless of the *p*-MSB concentrations. Furthermore, as shown in Appendix A, the absorption coefficient near the band edge was modeled by using Elliot’s theory of Wannier excitons [34] to assign the contributions from the excitonic peak (blue dashed line) and the lowest band of the continuum transition (green dashed line), respectively [35]. As such, we are able to extract the continuum absorption onset energy, that is, the bandgap energy *E*_g_ ≈ 2.200 eV, as well as the center of exciton resonance *E*_0_ ≈ 2.192 eV. The exciton binding energy *R*_ex_ ≈ 7.911 meV is then determined by their difference: *R*_ex_ = *E*_g_ − *E*_0_. The room temperature photoluminescence (PL) spectra for each sample were recorded with 350 nm light excitation as shown in Figure 2b, and the elevated intensity of excitonic emission at 526 nm was constantly witnessed with the increased *p*-MSB contents from 0 to 1 mg/mL. Meanwhile, the emission peaks at ~422 and ~446 nm gradually became noticeable with the increased *p*-MSB concentrations, which can be ascribed to the stimulated emission from *p*-MSB NPs, as evidenced in Appendix A [36]. Accordingly, the photoluminescence quantum yield (PLQY) of the as-synthesized 0D–2D heterostructure solution obviously developed as a function of *p*-MSB concentrations, resulting in a 70% increase from 30% to 51%, as shown in Figure 2c. Subsequently, the PLQY of the sample blending with 2 mg/mL *p*-MSB NPs was slightly decreased originating from the saturation of enhanced PL due to the reduced concentration of CsPbBr_3_ QDs. With the increased *p*-MSB concentrations, as shown in Figure 2d, the Commission International de L’Eclairage (CIE) chromaticity coordinates slightly developed from (0.156, 0.782) to (0.163, 0.670) within the green region [37], which can further confirm the neglectable effect on color variation with the incorporation of *p*-MSB NPs.

The CsPbBr_3_ QD/*p*-MSB NP thin films were fabricated via spin-coating of the 0D–2D heterostructure solutions with a variation of *p*-MSB concentrations. As clearly shown in Figure 3a–g, the surface morphologies appeared comparable throughout the whole concentration variations, verifying a feasible approach for the fabrication of additional strain-free thin films with enhanced light-emission efficiency. As a result, a comparable root-mean-squared roughness (R_RMS_) of ~44 nm was evenly witnessed with samples as shown in Appendix A. Regardless of the *p*-MSB concentrations, the uniform elementary distribution of Cs, Pb, and Br was observed without any localized aggregations, as revealed in Figure 3h and Appendix A. Meanwhile, the additional cyan bright spots appeared evenly on the thin films with the increased *p*-MSB concentrations under UV light illumination, as shown in the optical pictures in Figure 3i, which further evidenced the homogenous distribution of *p*-MSB NPs. As shown in Appendix A, the similar atomic ratios were obtained for samples Sp2 (Cs:Pb:Br = 0.59:1:1.85) and Sp7 (Cs:Pb:Br = 0.53:1:1.87), which were in accordance with the stoichiometry of the CsPbBr_3_ thin films in previous works [38]. The X-ray-diffraction (XRD) patterns of the samples are presented in Figure 3j, and the peaks appearing at 15.1°, 21.4°, 30.5°, 33.9°, and 37.7° can be assigned to the (100), (110), (200), (210), and (211) planes for the cubic phase of CsPbBr_3_, respectively [39]. The characteristic diffraction peak of the (006) plane for *p*-MSB was observed at 13.6° for the samples with relatively higher *p*-MSB concentrations [40,41]. To further explore the chemistry of *p*-MSB NPs and heterostructure samples, the X-ray photoelectron spectroscopy (XPS) analyses were carried out. The results in Appendix A show that the peak located at 283 eV corresponds to the C 1s of *p*-MSB, while the C 1s peak of Sp7 is located at 284 eV, consistent with previous reports [42]. The position of the C 1s peak in the Sp2 sample is consistent with that in *p*-MSB, suggesting that the composites possess different chemical states with CsPbBr_3_ QDs [43]. The Raman spectra of Sp2, Sp7, and *p*-MSB are compared in Appendix A. The Sp7 shows characteristic peak at 64.7 cm^−1^, which is consistent with previous research [43]. In the case of Sp2, this peak barely shifts, indicating there is no effect of *p*-MSB on the Raman spectra [44].The evolution of optical properties of the 0D–2D heterostructure thin films as a function of *p*-MSB concentrations is shown in Figure 4. As clearly revealed in Figure 4a, the excitation absorption peak of each sample was similarly observed at 514 nm, which differed from the characteristic peak of *p*-MSB thin films at ~400 nm as shown in Appendix A. An ignorable effect on the crystallinity of CsPbBr_3_ thin films by blending with *p*-MSB NPs is suggested, eventually resulting in a comparable optical bandgap of ~2.36 eV for each sample, as shown in Appendix A. Appendix A describes the contribution of free carriers and excitons to the absorption, which is separated by fitting the experimental data using the Elliott equation [34,45]. By this method, we extracted the exciton resonance (*E*_0_) and bandgap energy (*E*_g_) that are centered at 2.184 and 2.199 eV, respectively. This renders the exciton binding energy (*R*_ex_) to be 15.431 meV. As shown in Figure 4b, the intensity of the emission peak at ~526 nm noticeably elevated with the increasing contents of *p*-MSB, and the emission peaks that occurred at ~461 and ~490 nm can be ascribed to the contribution of the *p*-MSB NPs, as evidenced in Appendix A. As shown in Figure 4c, the typical type-II heterostructure [46] can be obtained due to the relatively larger bandgap (E_g_) of *p*-MSB (2.9 eV) [44] than that of CsPbBr_3_ (2.3 eV) [47]. Under light illumination, the spontaneous transfer of photogenerated electrons was correspondingly induced by the potential difference between *p*-MSB (conduct band, E_C_ = 2.7 V) [36] and CsPbBr_3_ (E_C_ = 3.6 V) [48] and the holes collected from CsPbBr_3_ to *p*-MSB. Given that the electron mobility of *p*-MSB (0.13 cm^2^ V^−1^ s^−1^) [49] was much higher than the hole mobility of CsPbBr_3_ (0.01 cm^2^ V^−1^ s^−1^) [50], the intensified radiative combination can be expected within CsPbBr_3_, along with the injection of extra electrons. As a result, the PLQY sharply increased from 20% to 40% when the *p*-MSB concentration reached 1 mg/mL, and the saturation of the enhancement in PLQY was witnessed with further increased *p*-MSB concentration due to the dominant charge carrier separation driven by the built-in electric field as shown in Figure 4d. The charge carrier dynamics within the heterostructures were revealed with the time-resolved photoluminescence spectra (TRPL) in Figure 4e, which were fitted with a triexponential decay model by using the Levenberg–Marquardt iteration algorithm.

**Figure 3 nanomaterials-13-02723-f003:**
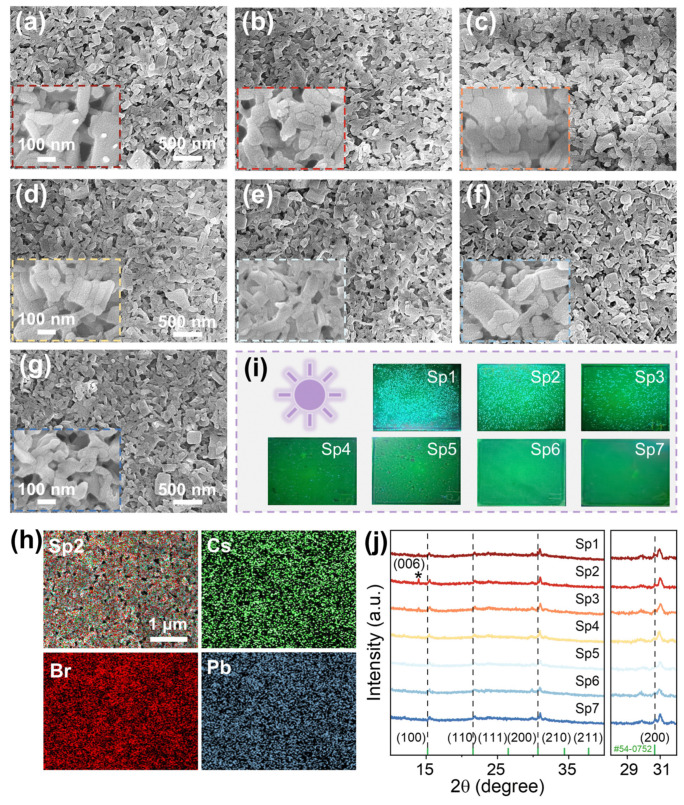
Scanning electron microscope (SEM) images of hybrid CsPbBr_3_ QD/*p*-MSB NP thin films with different *p*-MSB concentrations: (**a**) 2 mg/mL (Sp1), (**b**) 1 mg/mL (Sp2), (**c**) 0.5 mg/mL (Sp3), (**d**) 0.1 mg/mL (Sp4), (**e**) 0.05 mg/mL (Sp5), (**f**) 0.025 mg/mL (Sp6), (**g**) 0 mg/mL (Sp7). (**h**) Energy-dispersive X-ray spectroscopy (EDS) element maps of sample Sp2. (**i**) The optical pictures of hybrid CsPbBr_3_ QD/*p*-MSB NP samples under UV illumination. (**j**) X-ray diffraction (XRD) spectra of the hybrid CsPbBr_3_ QD/*p*-MSB NP thin films (* marks the position of peak (006)).


(2)
$$A=A_{1}e^{-t/\tau_{1}}+A_{2}e^{-t/\tau_{2}}+A_{3}e^{-t/\tau_{3}}$$


Here, *τ*_1_, *τ*_2_, and *τ*_3_ are related to the intrinsic radiative recombination (core state), surface-trap-assisted recombination (shallow level), and non-radiative Shockley–Read–Hall recombination (deep level), respectively [51]. *A*_1_, *A*_2_, and *A*_3_ are the corresponding intensities. The fast-decay components (short-lived lifetime) *τ*_1_ and *τ*_2_ can be related to the trap-assisted non-radiative recombination at the grain boundaries [52], and thus a shorter period was always witnessed with sample Sp2 (*τ*_1_ = 3.35 ns, *τ*_2_ = 13.44 ns) than that of sample Sp7 (*τ*_1_ = 47.61 ns, *τ*_2_ = 38.28 ns), suggesting severe defect-related exciton trapping behavior for the pristine CsPbBr_3_ QDs, as shown in Appendix A. Meanwhile, accelerated radiative recombination was evidenced with a smaller slow-decay component (long-lived lifetime) *τ*_3_ of sample Sp7 (5.74 ns) compared with sample Sp2 (6.28 ns) [53]. As revealed in Figure 4f, the chromaticity coordinates of the samples remained in the green region with a slight shift, which was consistent with the evolution witnessed in solutions.

**Figure 4 nanomaterials-13-02723-f004:**
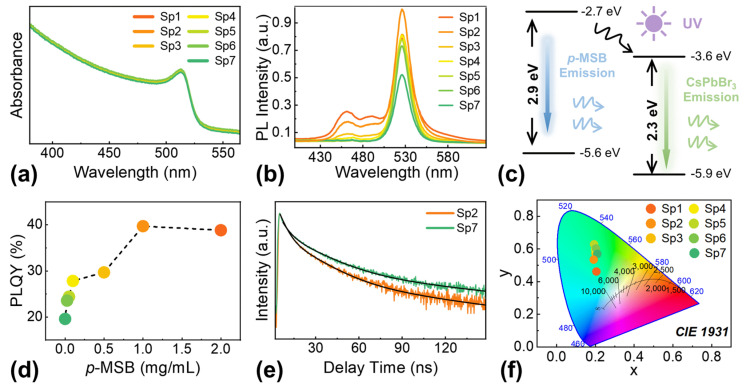
(**a**) Absorption spectra, (**b**) normalized PL spectra of the hybrid CsPbBr_3_ QD/*p*-MSB NP thin films. (**c**) The energy band diagram under light illumination in the CsPbBr_3_ QD/*p*-MSB NP heterostructures. (**d**) PLQY plots. (**e**) Time−resolved PL (TRPL) spectra of samples Sp2 and Sp7. (**f**) CIE coordinates of the hybrid CsPbBr_3_ QD/*p*-MSB NP thin films.

The thermal annealing effect on the optical and morphological properties of the 0D–2D heterostructure thin films with an identical *p*-MSB concentration of 1 mg/mL was evaluated by varying the temperatures between 25 and 100 °C. As shown in Appendix A, the grain aggregation of the 0D–2D CsPbBr_3_ QD/*p*-MSB NP thin films was gradually observed as a function of temperature, leading to an inclined R_RMS_ from 31.0 to 82.5 nm, as revealed in Appendix A. As observed with the XRD patterns in Figure 5a, the characteristic perovskite phase peaks along (100) and (200) slowly became noticeable with the variation in annealing temperatures, and the peak at 12.7° occurred at relatively higher temperatures, manifesting the formation of CsPb_2_Br_5_ [54,55]. As shown in Appendix A, the emission peak at 526 nm was evenly observed with the samples annealed at various temperatures, and the peak intensity gradually increased depending on temperature because of the improved crystallinity [56]. Nevertheless, the damping of PL emission was witnessed with the samples annealed above 60 °C, as shown in Appendix A, which can be possibly caused by the intensified non-radiative recombination along with the overgrowth of grains [57]. Therefore, the average lifetimes (τ_avg_) elongated from 14.15 to 18.97 ns with the elevated temperatures, proving the concurrently boosted recombination for the radiative and non-radiative processes, as shown in Figure 5b and Appendix A [53].

To further estimate the stability for practical applications, the performance of the 0D–2D CsPbBr_3_ QD/*p*-MSB NP thin films fabricated with 1 mg/mL *p*-MSB was methodically investigated under diverse conditions, as shown in Figure 5c–g. At room temperature, the sample exhibited excellent stability with a comparable intensity, even after 1 month exposure under an ambient atmosphere as shown in Appendix A, and the slight increase can be possibly attributed to the passivation of adsorbed gas molecules on the surface defects [58]. As shown in Appendix A, continuous post-heating at various temperatures was applied for samples Sp2 and Sp7 to simulate the different temperature working conditions, and the deterioration in PL emission intensity was observed within the whole temperature range between 25 °C and 65 °C. Different from the annealing process during fabrication, the gradual adsorption of gas molecules on the surface defects tended to boost the PL emission owing to the reduced non-radiative recombination centers [59], and the PL damping can be expected with the exposure of those defects along with the accelerated desorption of the gas molecules [60]. Figure 5c shows that the PL intensity of sample Sp2 decays less with the increase in temperature than that of sample Sp7. Compared with room temperature, the PL intensity remained 58% for sample Sp2 and 34% for sample Sp7 at an identical temperature of 65 °C, indicating the improved thermal stability with the existence of *p*-MSB NPs. It is evident that the *p*-MSB NPs on the surface of the CsPbBr_3_ QDs serves as a barrier for oxygen and can block ion migration of inter-particles, thus prohibiting the accompanying ripening growth and PL quenching of the CsPbBr_3_ QDs caused by heating [61]. To date, the chronical stability against ambient atmosphere is still a great challenge for perovskite optoelectronic devices due to the undesirable decomposition of perovskite with the corrosion of water molecules [62]. Figure 5d and Appendix A show the variations in the PL intensity of 0D–2D CsPbBr_3_ QD/*p*-MSB NP thin films under low RH (30%), middle RH (75%), and high RH (84%) conditions. The 30%, 75%, and 84% humidity environments are achieved by supersaturated salt solutions of LiF, NaCl, and KCl (25 °C) in a sealed chamber, respectively. As witnessed in Figure 5d, sample Sp2 generally exhibited relatively better stability at each condition owing to the improved stability with *p*-MSB NPs. Thus, the PL intensity of sample Sp2 remains over 74%, and that of sample Sp7 significantly declined by 66% at 75% RH for 12 h due to much more severe decomposition with the formation of by-products based on the Equation (3) [63] as shown in Appendix A.

**Figure 5 nanomaterials-13-02723-f005:**
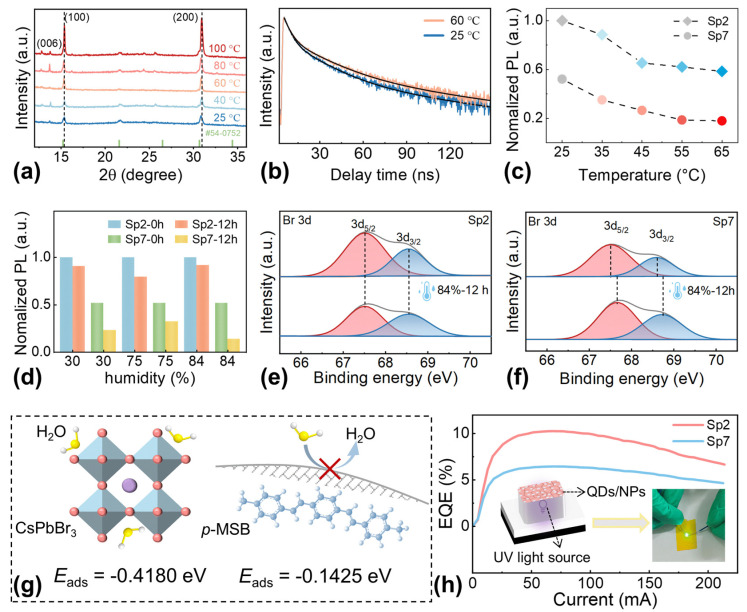
(**a**) XRD spectra and (**b**) TRPL spectra of the hybrid CsPbBr_3_ QD/*p*-MSB NP thin films blending with 1 mg/mL *p*-MSB at different annealing temperatures. (**c**) The PL intensities as a function of temperatures for samples Sp2 and Sp7. (**d**) PL intensities of samples Sp2 and Sp7 as fabricated and after storing for 12 h at 30% RH,75% RH, 84% RH, respectively. (**e**) X-ray photoelectron spectroscopy (XPS) spectra of Br 3d signatures on sample Sp2 as fabricated and after storing for 12 h at 84% RH. (**f**) X-ray photoelectron spectroscopy (XPS) spectra of Br 3d signatures on sample Sp7 as fabricated and after storing for 12 h at 84% RH. (**g**) The calculated adsorption energies (*E*_ads_) of H_2_O molecules for the hybrid CsPbBr_3_ QD/*p*-MSB NP heterostructures. (**h**) EQE curve of the Sp2 and Sp7−based LED. (Insets) The configuration and optical image of the LED.


(3)
$$7\mathrm{CsPbB}r_{3}+2H_{2}O⇌3\mathrm{CsP}b_{2}Br_{5}+Cs_{4}\mathrm{PbB}r_{6}\cdot 2H_{2}O$$


Regardless of the RH, a much more noticeable damping of PL intensity was always observed with sample Sp2 as a function of time in comparison with sample S7 as shown in Appendix A. For sample Sp7, the PL emission peak at 75% RH was continuously higher than 84% RH at each duration, and the opposite trend was observed with sample Sp2, as show in Appendix A. It can because the residual precursors and by-products can be possibly catalyzed into CsPbBr_3_ by water molecules under proper humidity based on the flowing reaction [63]:(4)$$\mathrm{CsP}b_{2}Br_{5}+\mathrm{CsBr}⇌2\mathrm{CsPbB}r_{3}$$
(5)$$Cs_{4}\mathrm{PbB}r_{6}+3\mathrm{PbB}r_{2}⇌4\mathrm{CsPbB}r_{3}$$
(6)$$3\mathrm{CsP}b_{2}Br_{5}+Cs_{4}\mathrm{PbB}r_{6}⇌7\mathrm{CsPbB}r_{3}$$

Therefore, the crystallinity of CsPbBr_3_ was improved by the accelerated ion diffusion of precursors and by-products with the increased RH from 30% to 75%, which can compensate for the damping of PL intensity as shown in Figure 5d [64]. However, the PL intensity of Sp2 at RH 75% was lower than that at RH 30%, which can be caused by the hindrance of ion diffusion with the existence of *p*-MSB. As a result, the PL intensity damping of Sp2 at RH 84% became more moderated than that at RH 75%. With further increased humidity, the decomposition tended to become dominant with the restricted reaction of residual precursors and by-products [65], resulting in a drastic PL intensity damping of Sp2 from 75% RH to 84% RH after 12 h. To verify the moisture effect, Figure 5e,f shows the comparison on the X-ray photoelectron spectroscopy (XPS) spectra of Br 3d signatures for samples Sp2 and Sp7 before and after storing at 84% RH for 12 h, and the Cs 3d and Pb 4f spectra are revealed in Appendix A. As shown in Figure 5e,f, the obvious shifts in the Br 3d_3/2_ peak at ~67.5 eV and Br 3d_5/2_ peak at ~68.5 eV [66] were concurrently observed with sample Sp7 at 84% RH for 12 h, and the similar behavior was equally witnessed for Pb 4f signatures, as shown in Appendix A, verifying the accelerated generation of derivative-phase CsPb_2_Br_5_ and Cs_4_PbBr_6_ in Sp7 [67]. Appendix A shows the full-range survey scan XPS spectra of samples Sp2 and Sp7 as fabricated and after storing for 12 h at 84% RH. Based on the first-principles density functional theory (DFT) calculations, as shown in Figure 5g, the adsorption energies (*E*_ads_) of the water molecule on the surfaces of CsPbBr_3_ (−0.4180 eV) were much lower than that of *p*-MSB (−0.1425 eV), and thus the hydrophobicity of *p*-MSB can effectively prevent the direct contact of CsPbBr_3_ and water molecules, resulting in improved stability for 0D–2D CsPbBr_3_ QD/*p*-MSB NP [68]. The TRPL in Figure 4e shows that the heterostructure has fewer non-radiative recombinations caused by defects compared to the original quantum dots. Figure 5e,f and Appendix A exhibit XPS spectra of Br (67.5 eV, 68.5 eV), Cs (723.8 eV, 737.7 eV) and Pb (137.5 eV, 142.3 eV) elements. Both Sp2 and Sp7 show no obvious peak transformation for Br 3d, Cs 3d, and Pb 4f, indicating no new chemical bonds formed between CsPbBr_3_ QDs and *p*-MSB NPs [69]. In other words, *p*-MSB NPs does not passivate CsPbBr_3_ QDs by replacing halide defect sites. Thus, *p*-MSB NPs may passivate defects by wrapping on the surface of quantum dots, as shown in Figure 1b, and then prevent the invasion of oxygen and water molecules. Finally, we spin-coated the 0D–2D CsPbBr_3_ QD/*p*-MSB NP with the optimized ratio of the UV light source (0603-0.55T, Shenzhen Pink Purple Industrial Co., Ltd., Shenzhen, China) for the fabrication of green LEDs. As shown in Figure 5h, a maximum EQE of 10.26% was obtained, which is higher than the EQE (6.45%) of CsPbBr_3_ QDs. As shown in the insets of Figure 5h, the bright-green color was observed by the LED fabricated with the 0D–2D CsPbBr_3_ QD/*p*-MSB NPs, indicating the great potential for the application of light sources.

## 4. Conclusions

In summary, we developed a facile method to prepare highly luminescent and ultra-stable 0D–2D CsPbBr_3_ QD/*p*-MSB NP heterostructures under atmospheric conditions. The typical type-II heterostructure spontaneously formed with the in situ nucleation of CsPbBr_3_ QDs on *p*-MSB NPs, and the PLQY drastically increased by 200% for a narrow emission band around 526 nm obtained by the heterostructure thin films with 1 mg/mL *p*-MSB than that of the pristine one. Meanwhile, the robust parclose screen constructed with *p*-MSB NPs successfully prevented CsPbBr_3_ QDs from attacking with oxygen and moisture, resulting in a comparable PL intensity of the heterostructure thin films under exposure in atmospheric conditions for 1 month. Correspondingly, compared with the pristine sample, relatively less decreases were observed with the heterostructure thin films after continuous post-heating at each temperature, indicating that the existence of *p*-MSB NPs retarded the absorption of gas molecules on the surface defects. Given that the *E*_ads_ of CsPbBr_3_ (−0.4180 eV) was much lower than that of *p*-MSB (−0.1425 eV), the PL intensity of the pristine sample severely decreased by 75% after 12 h, and the long-term stability was witnessed with the heterostructure thin films even at RH 84%. Therefore, the synergistic strategy can excellently realize the high PLQY and stability of a perovskite-based light source. To fulfill the practical applications, the green-emitting LED with a high EQE of 9.67% was fabricated based on these QDs/NPs, paving a path for practical applications in UWOC systems.

## Figures and Tables

**Figure 1 nanomaterials-13-02723-f001:**
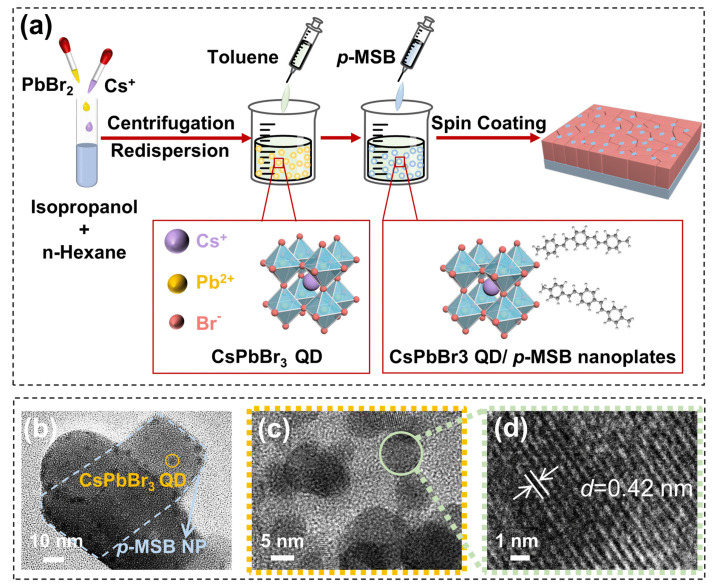
(**a**) Schematic illustration for the fabrication process of 0D−2D CsPbBr_3_ QD/*p*-MSB NP thin films. The transmission electron microscope (TEM) images of (**b**) CsPbBr_3_ QDs on *p*-MSB NP and (**c**,**d**) CsPbBr_3_ QDs at different magnifications.

**Figure 2 nanomaterials-13-02723-f002:**
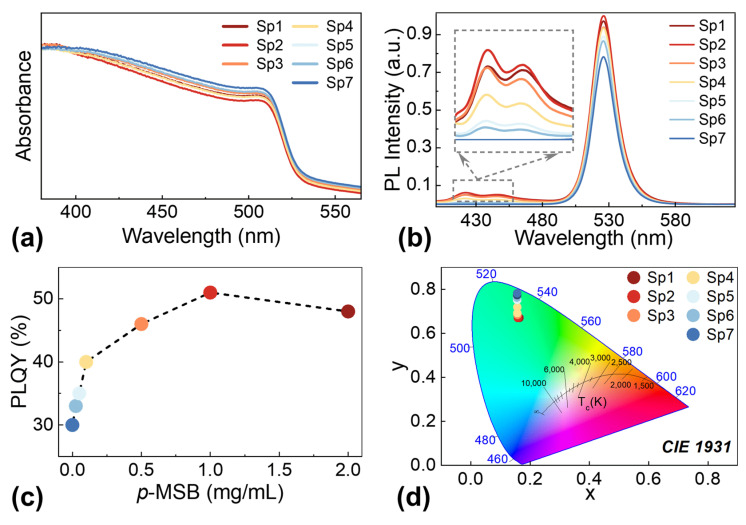
(**a**) Absorption spectra, (**b**) normalized photoluminescence (PL) spectra, (**c**) PLQY plots, and (**d**) CIE coordinates of hybrid CsPbBr_3_ QD/*p*-MSB NP solutions with various *p*-MSB concentrations: Sp1 (2 mg/mL), Sp2 (1 mg/mL), Sp3 (0.5 mg/mL), Sp4 (0.1 mg/mL), Sp5 (0.05 mg/mL), Sp6 (0.025 mg/mL), Sp7 (0 mg/mL).

## Data Availability

The data are available upon reasonable request from the corresponding author.

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
