# Peer review of "A Novel Strategy for the Synthesis of High Stability of Luminescent Zero Dimensional–Two Dimensional CsPbBr3 Quantum Dot/1,4-bis(4-methylstyryl)benzene Nanoplate Heterostructures at an Atmospheric Condition"

_nanomaterials, 2023, doi:10.3390/nano13192723_

Round 1
Reviewer 1 Report
I have read your manuscript entitled “A Novel Strategy for the Synthesis of High Stability of Luminescent 0D-2D CsPbBr3 QD/p-MSB Nanoplate Heterostructures at an Atmospheric Condition” and find it interesting and suitable for publication in Nanomaterials MDPI. Your work presents new heterostructures of perovskite QDs and organic NP. However, I believe that your manuscript still lacks some data to prove the existence of 1,4-bis(4-methylstyryl)benzene (p-MSB) NPs. Therefore, I recommend major revisions with some additional data expected for MSB NPs only:
- XPS: We need to know the chemistry of the material, contaminants, or quantifying dopant. XPS can provide this information.
- Raman: We need to understand the phase transitions of organic ligand in those perovskite heterostructures.
Besides that, there is no derivation of bandgap values from absorption spectra in Figs. 2 and 4. Elliot fit needs to be performed to see the contribution of free carriers and excitons to the absorption (see Yang et al, Nat. Comm. 7 12613 (2016). It is important to know the effect of MSB concentrations to free carriers and excitons. Also, there is no study of afterglow. With afterglow, you can know if MSB NP will induce defects or not. Therefore, without those data, it is difficult to accept this manuscript into Nanomaterials with significant high impact factor. Additionally, please add some publications with scintillator applications for the impact of your NPs to the QDs, such as Q. Chen, Nature 2018, 561, 88– 93 and F. Maddalena, ACS Appl. Mater. Interfaces 2021, 13, 49, 59450–59459.
Reviewer 2 Report
After carefully review the manuscript entitled: "A Novel Strategy for the Synthesis of High Stability of Luminescent 0D-2D CsPbBr3 QD/p-MSB Nanoplate Heterostructures at an Atmospheric Condition, I can mention that the paper is interesting and it highlights a kind of encapsulation where the CsPbBr3 are covered with p-MSB Nanoplates, preparing a heterostructure. I address some comments to be checked before to accept this manuscript for publication in Nanomaterials:
1. What is the passivation mechanism where the p-MSB nanoplates fill/replace halide defect sites on the perovskite surface? This could be discussed since the surface chemistry of the perovskite QDs is very labile.
2. In Figure 5h, page 9, I recommend to incorporate the EQE of the perovskite reference to observe the differences with the modified nanocomposite.
3. From equation 3 to 6, there are some errors in the subscripts. Please correct them.
4. Stability measurements in presence of O2/H2O or some of them should be described in the text, in order to provide to the readers, the main feature of the p-MSB coverage to offer a protection effect on the structural integrity of the QDs.
Minor errors can be addressed along the main text.
Round 2
Reviewer 1 Report
Thanks for addressing my comments. Accept as it is :)
Reviewer 2 Report
The paper can be publishable in the current form.
There are not english errors.